# Gut Microbiome Composition and Variance Are Modified by Degree of Growth Failure in Preterm Infants: A Prospective Study

**DOI:** 10.3390/nu17243907

**Published:** 2025-12-13

**Authors:** Katherine A. Stumpf, Miranda Green, Xinying Niu, Dongmei Lu, Shuheng Gan, Xiaowei Zhan, Maricel N. Maxey, Monica Boren, Sujir Pritha Nayak, Sana Jaleel, L. Steven Brown, Jane A. Foster, Julie Mirpuri

**Affiliations:** 1Division of Neonatal-Perinatal Medicine, Department of Pediatrics, University of Texas Southwestern Medical Center, Dallas, TX 75390, USA; xinying.niu@utsouthwestern.edu (X.N.); dongmei.lu@utsouthwestern.edu (D.L.); pritha.nayak@utsouthwestern.edu (S.P.N.); sana.jaleel@utsouthwestern.edu (S.J.); 2Department of Psychiatry & Behavioral Neuroscience, McMaster University, Hamilton, ON L8S 4L8, Canadajane.foster@utsouthwestern.edu (J.A.F.); 3Quantitative Biomedical Research Center, Peter O’Donnell Jr. School of Public Health, University of Texas Southwestern Medical Center, Dallas, TX 75390, USA; shuheng.gan@utsouthwestern.edu; 4Center for Genetics of Host Defense, School of Public Health, University of Texas Southwestern, Dallas, TX 75390, USA; xiaowei.zhan@utsouthwestern.edu; 5Parkland Health, Dallas, TX 75235, USA; maricel.maxey@phhs.org (M.N.M.); monica.boren@phhs.org (M.B.); steven.brown@phhs.org (L.S.B.); 6Center for Depression Research and Clinical Care, Department of Psychiatry, O’Donnell Brain Institute, University of Texas Southwestern Medical Center, Dallas, TX 75390, USA

**Keywords:** gut microbiome, preterm, neonatal, growth, neonate, feces, 16s rRNA

## Abstract

**Background/Objectives**: Preterm infants often require increased caloric intake to maintain appropriate growth while in the neonatal intensive care unit (NICU). Emerging evidence suggests that alterations of the gut microbiome may play a role in infant and childhood growth patterns. The fecal microbiome patterns in infants with normal and poor growth patterns were classified in this study. **Methods**: We conducted a prospective trial of infants of less than 29 weeks’ gestation with an embedded case–control analysis of infants with normal or poor growth patterns. Fecal samples were collected weekly from infants on full enteral feeds and analyzed blindly using 16s rRNA next-generation sequencing. The relationship between gut microbial diversity and composition and growth pattern and trajectory were assessed. **Results**: A total of 115 infants were enrolled in the trial with 263 fecal samples selected from 87 enrolled infants for analysis. In total, 37 samples were available from the normal growth cohort, 56 samples from the poor growth cohort, and 170 samples were available for analysis from the very poor growth cohort. Analysis of relative abundance revealed increased representation of *Veillonella*, *Bifidobacterium*, and *Clostridium* in very poor growth infants compared to normal growth infants. Variation in specific taxa was also found to vary significantly across post-menstrual age depending on the degree of growth failure. **Conclusions:** Gut microbiome composition and variance was modified by the degree of growth failure in our cohort of preterm infants. Our study adds to the growing body of evidence that alteration of the microbiome is associated with poor growth in preterm infants. This may ultimately represent a therapeutic target for growth failure in preterm infants.

## 1. Introduction

Preterm infants require adequate growth to achieve optimal outcomes including neurodevelopmental achievements [1,2,3]. About half of all premature infants develop growth failure, often without a clear etiology [4]. While it is recommended that preterm infants grow at a rate similar to expected fetal growth, this is rarely achievable [5]. Additionally, nutritional needs vary widely from infant to infant, with most preterm infants requiring 100–150 kcal/kg/day to achieve reasonable growth [6].

Emerging evidence suggests that growth failure in children may be associated with gut microbiome composition [7]. Blanton et al. demonstrated that transplantation of a poor growth microbiome from malnourished children to a mouse model led to corresponding phenotypic changes, which could be partially corrected by the introduction of a “healthy” microbiome [8]. The preterm microbiome is affected by numerous factors including exposure to antibiotics, mode of delivery, and feeding substrates [9]. Younge et al. demonstrated that a disruption of the preterm microbiome may be linked to growth failure in preterm infants [10].

In a systematic review, Neves et al. described mechanistic models which may serve as causal links between growth and the intestinal microbiome [4]. This includes the fermentation of non-digestible substrates by microbes to create absorbable energy as noted in several animal models [11,12]. Additionally, the importance of gut bacteria interacting with human milk oligosaccharides is described and demonstrated in Charbonneau’s piglet/mouse models which showed increased weight gain, bone mineral density, and cortical thickness [13].

The association of the intestinal microbiome of preterm infants with growth failure is not fully understood, as studies in this complex problem remain limited. In particular, the difference between infants with high caloric needs to maintain growth compared to those with normal caloric needs has not been described in a prospective cohort. We hypothesized that specific gut microbial patterns would be associated with poor growth in preterm infants with high caloric intake compared to those with appropriate growth on normal caloric intake. This study provides longitudinal microbiome data from a large preterm infant population in an urban neonatal intensive care unit (NICU).

## 2. Materials and Methods

### 2.1. Patients and Study Design

In this nested case–control study within a prospective cohort, we enrolled 115 infants in the primary cohort of less than 29 weeks’ gestation at birth and classified them based on postnatal growth outcomes. Enrollment was performed by parental informed consent using an IRB-approved consent form in the primary language of the participant. The study included infants who were born at Parkland Health and admitted to the NICU after birth between June 2019 and August 2022. Sample size was based on the assumption that the normal growth group would have an alpha diversity mean of 100 and the other groups would have a difference of 20% with a common standard deviation equal to 20. Based on the null hypothesis of no difference in alpha diversity between groups, a total of 108 infants achieves 90% power to detect differences among means versus the alternative of equal means using an F test with a 0.05 significance level. The study included infants who were born at Parkland Health and admitted to the NICU after birth between June 2019 and August 2022. Stool samples were collected weekly from infants receiving “full volume” feeds, which was defined as receiving a minimum of 120 kcal/kg/day without IV nutrition in place. All infants in this study were receiving full feeds without IV nutrition at the time of stool collection. No probiotics, prebiotics, or lactoferrin were administered to any infant. Infants with major congenital anomalies expected to affect growth were excluded from the trial.

Growth parameters were evaluated weekly by unit dieticians, with desired and actual weight gain documented in the electronic medical records. We structured cohorts of infants according to weight gain, including those reaching expected dietician goal gains, who were defined as normal, or the control group (N); those with poor growth (PG), who were within 1–20% below the expected weight gain for the week; and a very poor growth cohort (VPG), which consisted of infants > 20% below their expected weight gain for the week despite receiving optimized caloric intake (cases). Each infant was evaluated weekly during the study for expected and actual weight gain, providing longitudinal data.

Infant and maternal demographic data, infant characteristics, major diagnoses, feeding substrates, infectious data including culture results, and use of antibiotic therapy were collected for each enrolled infant. Statistical analysis of the enrolled population and growth cohorts was completed using ANOVA and Chi-square testing.

### 2.2. Sample Collection and Isolation of Fecal Genomic DNA

Between 1 and 16 fresh fecal samples were collected from each preterm infant. Sample collection was standardized with the collection of fecal samples immediately after diaper checks in the neonatal intensive care unit. Fresh fecal samples were stored at −80 °C for further analysis. Fecal specimens were lysed by bead-beating (0.1 mm zirconia/silica beads) and subjected to additional phenol/chloroform extractions. DNA concentrations were quantified by fluorescence-based assay (Quant-iT PicoGreen dsDNA, Life Technologies, Carlsbad, CA, USA). From each sample, 16S rRNA genes (variable region 4, V4) were amplified using uniquely bar-coded primers. Polymerase chain reaction (PCR) consisted of Accuprime Pfx Supermix, primers, and a template. Following amplification, PCR products were verified, cleaned, and normalized. Pooled samples were sequenced using Illumina MiSeq (PE-250, San Diego, CA, USA).

### 2.3. Consent

This study was approved by the University of Texas Southwestern Medical Center and Parkland Hospital Institutional Review Boards (Dallas, TX, USA), and infants were enrolled after parental written consent was given. IRB approval number: STU 042018-061.

### 2.4. Statistical Analysis of Microbiome Data

#### 2.4.1. Alpha and Beta Diversity Analysis

All downstream analyses of preprocessed microbiome data were performed in R version 4.3.3. Alpha diversity (Shannon, Inverse Simpson, and Observed reads/richness) and total sequencing depth were calculated for each sample; samples with a Shannon diversity or read count of less than 2 standard deviations below the grand mean were excluded from subsequent assessment (Shannon: 1.445–2 × 0.560, final cutoff of x > 0.323; read count: 25,993.2–2 × 9340.8, final cutoff x > 7311). Trends in alpha diversity across the postnatal period were assessed via a general linear mixed model (GLMM) accounting for clinical covariates, growth status, and within-infant repeated measures; additional post hoc tests were performed to confirm pairwise differences in significant covariates with more than two levels. The effect size of clinical covariates was assessed via the R2 derived from PERMANOVA (adonis2 function; vegan R package); variance explained by each covariate was examined via distance-based redundancy analysis (db-RDA via the capscale function, also from the vegan package) on Aitchison and Bray–Curtis distance metrics. The first 5 principal coordinates (PCs) of distance matrix decompositions were correlated to clinical covariates using the rcorr package with additional BH *p*-value correction.

Differential Abundance Analysis: Prior to differential abundance (DA) analysis, raw microbiome data was denoised and filtered using our retain-resolve (RR) strategy as previously described [14,15]. Out of 506 initial taxa, 19 amplicon sequence variants (ASVs) met the initial prevalence/abundance criteria and were retained in the analytical dataset. The remaining 487 ASVs that did not meet the criteria were agglomerated to 140 genus-level taxa using the tax glom function. A total of 11 glommed taxa met the criteria in the second round of filtering and were added to the analytical dataset. The remaining 129 taxa were agglomerated into a single “other” category, resulting in a final dataset that included a total of 31 taxa. Differential abundance analysis of taxa related to infant growth trajectory was limited to infants of average gestational age with at least 2 longitudinal samples, for a total of 165 observations across 51 infants across 3 brackets of growth percentile as defined previously. The main DA model was constructed using the linear regression for differential abundance analysis (LinDA) algorithm that accounts for the compositional nature of microbiome data and implements bias correction to compensate for compositional effects. Using standard lme4 syntax, the model formula was constructed as follows:X ~ growth%ile + antibiotic status + feeding group + mode of delivery + gestational age at birth + (1|individual ID) + (0 + day of life at sample|individual ID)
where infant ID is modeled as a random effect; the first term (1|individual) adds the scalar term with a random intercept for each individual, whereas the second term (0 + day of life_at_sample|individual) models the effect of the postnatal day within each subject. Significant taxa were defined as having a BH-corrected *p*-value < 0.05 and a mean absolute log^2^fold change > 1 (equivalent to doubled or halved mean abundance between groups). Significant taxa were further assessed for outliers and influence points via boxplots and scatter plots.

#### 2.4.2. Clustering Analysis

Taxonomic drivers of neonate gut microbial variability were explored via unsupervised clustering of Bray–Curtis dissimilarity measures with an agglomerative nesting algorithm (agnes). The cluster number was tuned using optimized within-sum-of-squares (WSS) and bootstrapped gap statistics. Association of individual clusters with categorical metadata was assessed using a Chi-squared test, while differences in continuous covariates between clusters were assessed using a Kruskal–Wallis test with *p*-value correction. Transitions of individual infants between clusters was visually assessed using the galluvial package.

## 3. Results

### 3.1. Infant Characteristics

A total of 115 infants were enrolled; 108 infants had stool samples collected during their NICU stay and 7 enrolled infants did not reach full feeds or were transferred to another facility prior to being eligible for stool collection. Demographic and baseline characteristics of all enrolled infants appear in Table 1. The mean birth weight (BW) and gestational age (GA) of enrolled infants was 971 g (range: 450–1700 g) and 26 + 6 weeks (range: 22 + 6–28 + 6), respectively. We collected over 800 stool samples and categorized each sample based on infant growth parameters at the timing of sample. Samples not meeting the criteria for each cohort were excluded from this analysis.

Samples were then cohorted according to growth status per week of the sample. A total of 37 samples meeting normal growth criteria, 56 samples meeting poor growth criteria, and 170 samples meeting very poor growth criteria were sent for next-generation sequencing and analyzed. Characteristics of the infants providing these samples at the time of sample collection appear in Table 2. Samples from infants in the normal growth cohort were more likely from infants of lower gestational age (26 weeks vs. 27 weeks, *p* < 0.05) and were more likely to be from singleton infants (*p* < 0.05). There was no difference in samples taken from infants pertaining to differences in gender, race, ethnicity, birth weight, mode of delivery, IUGR status, or in any common major diagnoses found in the neonatal population such as IVH, ROP, NEC, or BPD. Feeding regimen and antibiotic exposure were not statistically significant different between the growth groups (Table 2).

### 3.2. Microbiome Analysis

Out of 263 samples analyzed, 24 were initially excluded based on low alpha diversity and 21 samples based on a lack of longitudinal sampling (i.e., only one sample for a patient was collected) (Appendix A).

The remaining 218 microbiome samples spanned 28–41 weeks post-menstrual age (PMA) at the time of collection, which displayed a median sequencing depth of 24,687 reads, a minimum of 7321 reads, and a maximum of 74,155 reads. Compositional bar charts of microbiota at different taxonomic ranks and averaged across different experimental groups are shown in Figure 1A. In the overall sample, the top three taxa at the phylum level were *Proteobacteria* (58.8%), Firmicutes (37.2%), and *Actinobacteriota* (3.14%). At the family level, three dominant bacterial groups were identified: *Enterobacteriaceae* (55.8%), *Veillonellaceae* (14.9%), and *Enterococcaceae* (11.3%). Roughly 40% of 506 ASVs were not identified at the genus level; those classified to this resolution belonged primarily to the genera *Escherichia-Shigella* (27.6%), *Veillonella* (22.5%), *Enterococcus* (19%), *Clostridium sensu stricto* (4.8%), and *Bifidobacterium* (4.6%).

### 3.3. Growth Status and Individual Differences Contribute to Distinct Components of Neonate Microbiome Variability and Development

To assess the relationship between infant growth, clinical characteristics, and the microbiota, we first assessed differences in alpha and beta diversity between samples. Based on Bray–Curtis and Aitchison distances, infant microbiome composition was not systematically impacted by growth percentile (Figure 1B) or postnatal age (PND, Figure 1C). This was further supported by the PERMANOVA results, which revealed that individual infant ID contributed roughly 50% of overall variance in gut microbiome composition across all distance measures (Figure 1D). Inter-individual differences were followed by growth status, gestational age, weight, and feeding method, which each consistently contributed 1–2% of overall variance (Figure 1D). In addition, correlation analysis revealed no strong relationships between principal coordinate axes and clinical variables. Linear mixed models of alpha diversity metrics suggested a significant association between Shannon diversity and antibiotic treatment (B = −0.609, *p* < 0.05, Figure 1E) as well as post-menstrual age (PMA, B = 0.02, *p* < 0.001), indicating that infants receiving additional antibiotics had lower overall alpha diversity, while there was a modest but steady increase in diversity over time across all infants. To identify whether these trends varied between different infant groups, we constructed an additional model that included interaction terms for growth + PMA, antibiotic status (Abx) + PMA, and mode of delivery + PMA. We found a strong interaction effect between PMA and two out of three variables tested (Figure 1F,G); samples in the normal and poor growth (PG) category showed no significant association between PMA and Shannon diversity, while the very poor growth (VPG) infants showed a marked positive trend in both Shannon and inverse Simpson indices with increasing PMA (Figure 1E). Indeed, VPG samples showed a modest but significant positive correlation between age and alpha diversity metrics (inverse Simpson: r = 0.36, Shannon: r = 0.41, both *p* < 0.001) that was not observed in low growth or normal growth samples (r < 0.1, *p* > 0.1). A similar pattern was observed when comparing infants delivered either vaginally (vaginal birth, VB) or by cesarian section (CS), with the latter group displaying a stronger positive association between PMA and microbial richness (Figure 1G). To further investigate this interplay between alpha diversity and delivery mode, we compared the slopes of within-individual lines of best fit for infants in the VB and CS groups (Appendix A). We found that CS infants displayed a slight increase in regression slopes compared to VB infants, suggesting a positive temporal trend present in this group consistent with the majority of individual microbiota dynamics. Overall, these results suggest that while the composition of the preterm microbiota is largely determined by individual factors, clinical and environmental variables can systemically impact the longitudinal diversity of the microbiota, with implications for the growth status of the infant.

### 3.4. Keystone Taxa in the Infant Microbiome Show Different Temporal Dynamics in Relation to Preterm Growth Status

To further examine the dynamic relationship between microbiota and infant growth status, we conducted a differential abundance analysis within the subgroup of infants that were born average for gestational age (AGA, n = 165 samples across 51 infants). We constructed two models, one of which categorized infant samples according to growth bracket (PG or VPG), while the second included growth percentile as a continuous fixed effect. In both cases, antibiotic use, mode of feeding, mode of delivery, and gestational age at birth were included as fixed-effect covariates, while random intercepts and slopes for each taxa across each postnatal day were established as random effects.

Out of 30 RR ASVs analyzed, the first model identified only 8 showing significantly altered abundance across growth groups. Two Veillonella ASVs, two Bifidobacterium ASVs, one Clostridium ASV, and an additional ASV identified as Streptococcus salivarius showed a positive fold change in VPG infants compared to NG infants, while Negativicoccus succinicivorans was decreased in VPG infants and a single unidentified Staphylococcus ASV was decreased in PG infants (Figure 2A,B). Similarly, the second model found that three Veillonella ASVs were negatively associated with growth, including one (sp313) that was significantly increased in the VPG group (Figure 2A,B). The temporal dynamics of these differentially abundant communities were altered in PG and VPG infants (Figure 2C). For example, taxa such as Bifidobacterium ASV257 and Veillonella ASV307 were effectively absent in NG infants but showed consistent fluctuations in VPG samples across the perinatal period. By tracking infants across the measurement period, we found that mean fluctuations in taxon abundance were not due to disparate prevalence between individuals or driven by different individuals with or without a given microbe within the same growth bracket. Rather, many of these DA taxa showed significant fluctuations in abundance within the same individual—that is, infants would frequently transition between high and low abundance of the same taxon across sampling time points (Appendix A). Although it remains unclear whether this phenomenon is related to technical bias in microbial sequencing that may alter detection probability, these results suggest that the preterm infant microbiome is highly volatile even at the level of keystone taxa related to growth outcomes.

### 3.5. The Preterm Microbiota Fluctuates Between Multiple Steady States Across Early Life

To understand the bacterial taxa driving variability in the neonate microbiome, we performed hierarchical clustering on the Bray–Curtis distance matrix for both raw and RR microbiome data. This method identified four stable clusters, which was agreed upon by both WSS and GS optimization in both datasets (Figure 3A,B), suggesting that the ~30 unique taxa identified through the RR method are the main drivers of neonate GM variability. Examining the community composition of these clusters, a striking pattern emerged: clusters 1 and 2 were dominated by distinct *Enterobacteriaceae* ASVs, while cluster 3 was dominated by *Escherichia* and *Shigella*, and cluster 4 was co-dominated by the principal *Enterobacteriaceae* ASVs from cluster 1 and an additional *Enterococcus* ASV (Figure 3C).

Curiously, while some infants retained cluster affiliation across the study period, many changed cluster memberships up to 3 times across the postnatal window (Figure 3D). Clusters did not differ significantly in growth percentile, post-menstrual age, or postnatal age (Figure 3E) and were independent from all other continuous and categorical metadata (Kruskal–Wallis and Chi-squared tests, *p* > 0.05). Therefore, this data suggests that individualized and somewhat idiosyncratic dynamics in the perinatal microbiota are the main drivers of variability in the premature gut ecosystem. However, fluctuations between dominant taxa over time may translate into dynamic phenotypic differences in growth and development. In agreement with differential abundance results, it suggests that multiple transitions between microbiome steady states, characterized by specific dominant taxa, may play an integral role in determining the individual’s growth trajectory.

## 4. Discussion

We followed the intestinal microbiome of 115 preterm infants longitudinally from the time of full feeding volume until discharge or transfer to a higher level of care. Our results showed no differences in alpha or beta diversity as assessed by multiple testing modalities. Individual infant characteristics were the primary contributor to the composition of the microbiome, with additional variance due to growth status, gestational age, weight, and feeding method. Interestingly, we identified four unique microbiome clusters that were retained both across and within individual infants.

When analyzing the overall microbiome composition across normal, poor growth, and very poor growth samples, we did not find a difference by principal component analysis; however, we were able to identify differences in specific taxa. Important significant differences were found in the prevalence of *Veillonella*, *Bifidobacterium*, and *Clostridium*, which were increased in the very poor growth cohort compared to the normal growth cohort. These specific taxa are generally associated with improved energy harvesting and short-chain fatty acid production, which should be beneficial. The increase in these taxa in the poor growth group could represent either compensation of the microbiome in a metabolically poor environment or highlight the differences in the metabolic potential of the species within these taxa. Specifically, both the Firmicutes and Bacteroides species show significant genus-level differences in the degradation of carbohydrates, protein, and lipids [16]. *Negativococcus* was decreased in the very poor growth cohort. Previous studies have noted greater abundance of *Veillonellaceae* in an appropriate growth group and increased abundance of other bacteria including *Staphylococcaceae* and *Enterobacteriaceae* [10].

Multiple studies have confirmed that antibiotic treatment alters the microbiome of preterm infants [17,18]. We did see a significant difference in microbiome composition in relation to antibiotic treatment, though we saw no growth differences which could be attributed to antibiotics, since days of exposure did not correlate with growth cohorts [17,18]. This is consistent with a study by Pyle et al., who demonstrated that very-low-birth weight infants’ weight differences were not attributable to differences in length of antibiotic exposure [19]. It is interesting to note that the individual contributed most to the differences in the microbial composition across samples, with clinical and environmental variables impacting the longitudinal diversity of the microbiota with implications for growth status of the infant. Differences between the microbiome of infants delivered vaginally versus C-section were noted, which is also a well-studied phenomenon [20]. However, the growth differences could not be attributed to the mode of delivery.

The compositional differences across studies may suggest that local, environmental, and individual genetic factors may play an important role in the microbiome composition in preterm infants. A surprising but important finding of our study was the identification of unique microbiome clusters that were maintained within and among samples, which suggests that analysis of microbiome cluster shifts may be more important in determining growth potential. This makes physiological sense with the supporting evidence that community compositions remain stable due to competition and local factors. Other studies have also identified communities that are important in growth [7]. This may also point to the importance of the metabolic potential of these clusters, with future analysis of the metagenome and metabolites that are produced by these stable communities. This may be more important in directing growth potential in this population.

Another interesting finding was the fluctuation in specific taxa seen primarily in the very poor growth group, with the normal growth group showing minimal fluctuation in the Bifidobacterium, Clostridium and Veillonella species. This is a novel finding, and the mechanisms still require further investigation; however, local intestinal factors or diet composition such as fortification variation could play a role. This could result in a more dynamic composition of bacterial taxa over time and warrants further investigation.

## 5. Limitations

Our study was limited by several factors including the inability to follow some infants who were transferred to a higher level of care for surgical needs, which can be included in future studies. Additionally, fecal samples could not consistently be collected on the exact dates of growth parameter determination due to a lack of infant stooling or study logistics, leading to extrapolation of data time points to the closest temporal relationship. This can be improved in future studies by utilizing a swab collection of stool, which is minimally invasive. The majority of our patient population received a combination of donor milk and mother’s milk, leading to some heterogeneity of causes for differences in microbiome findings. Patients may also have been affected by comorbid conditions such as bronchopulmonary dysplasia (BPD) or patent ductus arteriosus (PDA), which could theoretically alter their growth at baseline. These confounding factors are difficult to study. Further studies with a larger cohort can minimize the heterogeneity effect, and linking microbiome to post-operative states of growth would provide additional information not included in our investigation.

## 6. Conclusions

Microbial diversity in our study showed consistency with other studies in regard to alterations due to antibiotic exposure, postnatal age, and mode of delivery. The primary three taxa seen in our overall sample were *Proteobacteria*, Firmicutes, and *Actinobacteriota*. Infants in different growth cohorts had unique differences in particular taxa, including increased representation of *Veillonella*, consistent with other similar studies of preterm growth and microbiome. Further studies are needed to explore the relationship between growth, prematurity, and microbiome cohorts to allow for possible customized interventions and treatments for infants with growth failure.

## Figures and Tables

**Figure 1 nutrients-17-03907-f001:**
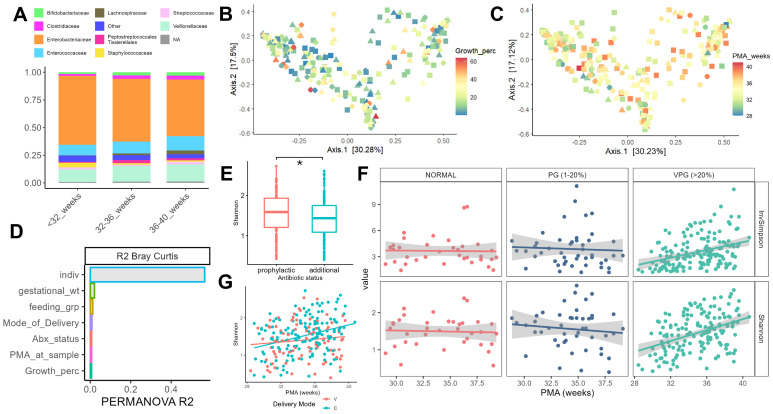
Microbiome diversity and composition of gut microbiome in preterm infants. (**A**) Compositional bar chart of the top 20 bacterial families constituting the infant microbiome at different post-menstrual age (PMA) brackets. (**B**,**C**) PCoA decomposition of Bray-Curtis distance matrix for infant microbiome dataset, colored by growth percentile (**left**) and PMA (**right**). (**D**) Results of PERMANOVA (999 permutations) on Bray-Curtis distance matrix, arranged according to R2 value (variance explained). (**E**) Shannon diversity differences between infants receiving prophylactic and additional antibiotics. (**G**) Relationship between PMA and Shannon diversity for infants born vaginally (V) or by cesarian section (C), lines of best fit show divergent trends across time. (**F**) Trends in Inverse Simpson (**top**) and Shannon (**bottom**) indices over time in normal (N), poor growth (PG) and very poor growth (VPG) samples. * *p* < 0.05.

**Figure 2 nutrients-17-03907-f002:**
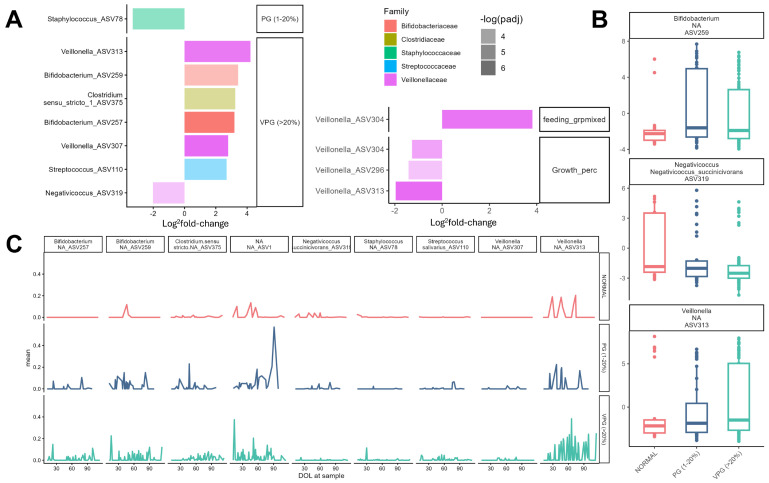
Differential Abundance Analysis of microbiome samples from infants across different growth trajectories. (**A**) Log-fold change of significant taxa (FDR < 0.05) identified as more/less abundant in low-growth samples compared to normal growth (**left**) and those positively/negatively associated with growth percentile (**right**). (**B**) CLR-transformed abundances of select DA taxa across infant growth brackets. (**C**) mean relative abundances of DA taxa by postnatal day, illustrating unique temporal fluctuations in taxon abundance across growth brackets.

**Figure 3 nutrients-17-03907-f003:**
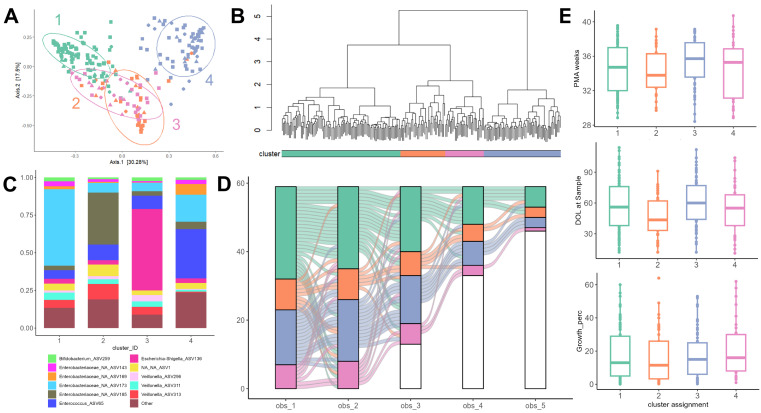
Hierarchical clustering of preterm infant microbiota into 4 unique community structures. (**A**) Bray-Curtis PCoA decomposition colored by assigned cluster. (**B**) Agnes dendrogram of microbiome samples, dashed line represents cut point used to define clusters. (**C**) Mean microbial composition of samples in each cluster, colored by RR-ASV. (**D**) Alluvial plot illustrating transitions of individual infants between cluster assignments across observations. Each connecting line represents the path of an individual infant across observations (obs). Most infants had <3 observations and therefore are not present in the 3rd, 4th, and 5th observation blocks (white bars). (**E**) distribution of other metadata including day of life (DOL), post-menstrual age (PMA), and growth percentile across clusters.

**Table 1 nutrients-17-03907-t001:** Baseline characteristics of all enrolled infants.

Gestational Age (weeks): mean (range)	26 + 6 (22 + 6–28 + 6)
Birth weight (grams): mean (range)	971 (450–1700)
AGA: n (%)	86 (80%)
SGA: n (%)	8 (7%)
LGA: n (%)	14(13%)
Gender: n (% male)	37 (32%)
Ethnicity: n (% Hispanic)	70 (65%)
Mode of Delivery: n (% C-section)	50 (46%)
Singleton: n (%)	82 (76%)
IUGR: n (%)	4 (4%)
Diagnoses during study total period:	
Necrotizing Enterocolitis (NEC); Any stage	4 (4%)
Bronchopulmonary Dysplasia (BPD); Any severity	80 (74%)
Patent Ductus arteriosus (PDA); Any severity	26 (24%)
Intraventricular Hemorrhage (IVH); Any grade	31 (29%)
% grade 3–4	4 (4%)
Retinopathy of Prematurity (ROP); Any stage ever	70 (65%)
% stage 3:	14 (13%)

**Table 2 nutrients-17-03907-t002:** Demographic and clinical characteristics of infants by growth cohort for stool samples selected and analyzed (calculated per sample).

	Normal Growth n = 37	Poor Growth n = 56	Very Poor Growth n = 170	*p* Value *
Gestational Age (weeks): mean ± standard deviation	26 ± 2	27 ± 2	27 ± 2	0.03
Male: n (%)	13 (35)	19 (34)	55 (33)	0.95
Race/Ethnicity: n (%)				0.57
White NH	3 (8)	1 (2)	6 (4)	
Black NH	11 (30)	19 (34)	42 (25)	
Hispanic	23 (62)	36 (64)	120 (71)	
Asian NH	0 (0)	0 (0)	1 (1)	
Birth weight (grams):				0.80 0.58
Mean ± standard deviation	957 ± 269	977 ± 253	950 ± 257	
AGA: n (%)	31 (84)	45 (80)	127 (75)	
SGA: n (%)	4 (11)	6 (11)	18 (11)	
LGA: n (%)	2 (5)	5 (9)	24 (14)	
C-section: n (%)	23 (62)	35 (63)	102 (60)	0.95
Singleton: n (%)	30 (81)	33 (59)	128 (76)	0.02
Intrauterine Growth Restriction (IUGR): n (%)	0 (0)	5 (9)	7 (4)	0.12
Feeding Regimen at time of Sample: n (%)				0.18
EBM/DBM + fortifier:	28 (76)	49 (88)	124 (73)	
EBM/DBM/Formula + fortifier:	1 (3)	2 (4)	14 (8)	
Formula only:	8 (22)	5 (9)	31 (18)	
Antibiotics given;				
At birth: n (%)	37 (100)	56 (100)	169 (100)	N/A
Within 7 days of sample:	4 (11)	1 (2)	6 (4)	0.08

* mean ± standard deviation compared with ANOVA test; n (%) compared with Chi-square test. AGA: appropriate for gestational age; SGA: small for gestational age; LGA: large for gestational age; IUGR: intrauterine growth restriction; EBM: expressed breast milk; DBM: donor breast milk; NH: non-Hispanic.

## Data Availability

The data that support the findings of this study are available from the corresponding author upon reasonable request due to patient privacy.

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
