# Peer review of "Gut Microbiome Composition and Variance Are Modified by Degree of Growth Failure in Preterm Infants: A Prospective Study"

_nutrients, 2025, doi:10.3390/nu17243907_

Round 1
Reviewer 1 Report
Comments and Suggestions for Authors
Journal
Nutrients (ISSN 2072-6643)
Journal
nutrients-4010918
Type
Article
Title
Gut microbiome composition and variance are modified by degree of growth failure in preterm infants
Authors
Katherine A Stumpf * , Miranda Green , Xinying Niu , Dongmei Liu , Shuheng Gan , Xiaowai Zhan , Maricel N. Maxey , Monica Boren , Sujir Pritha Nayak , Sana Jaleel , L. Steven Brown , Jane A. Foster , Julie Mirpuri *
Section
Nutrition and Metabolism
ABOUT THE MANUSCRIPT
- Based on the stamen that preterm infants often require increased caloric intake to main-tain appropriate growth while in the neonatal intensive care unit, the authors performed a prospective trial with infants less than weeks gestation with an embedded case control analysis of infants with normal or poor growth patterns. For this objective, they collected fecal samples on full enteral feeds and analyzed blindly using 16s rRNA next generation sequencing. Their results showed that the gut microbiome composition and variance were modified by the degree of growth failure.
- The authors did not include the number of pages in the manuscript.
TITLE
- The title of this manuscript is “Gut microbiome composition and variance are modified by degree of growth failure in preterm infants.
I suggest changing to “Gut microbiome composition and variance are modified by degree of growth failure in preterm infants: a prospective study”.
- It is important to include in the title the type of article
ABSTRACT
This section is adequate.
KEYWORDS
Keywords are:
“gut microbiome; preterm; neonatal; growth; neonate”
I suggest “
Keywords: gut microbiome; preterm; neonatal;; growth; feces;
16s rRNA”.
INTRODUCTION
- This section has merit but also some weaknesses. In these, we can say that the literature review could be more complete. I see the need to include more recent studies, primarily published in 2025.
There is only one reference published in 2025:
Li F, Hooi SL, Choo YM, Teh CSJ, Toh KY, Lim LWZ, et al. Progression of gut microbiome in preterm infants during the first three months. Sci Rep. 2025 Apr 9;15(1):12104.
Only one published in 2024:
(Fenton TR, Merlino Barr S, Elmrayed S, Alshaikh B. Expected and Desirable reterm and Small Infant Growth Patterns. Adv Nutr Bethesda Md. 2024 Jun;15(6):100220);
Only 3 published in 2023:
Neves LL, Hair AB, Preidis GA. A systematic review of associations between gut microbiota composition and growth failure in preterm neonates. Gut Microbes. 2023 Dec 31;15(1):2190301;
Xiu W, Lin J, Hu Y, Tang H, Wu S, Yang C. Assessing multiple factors affecting the gut microbiome structure of very preterm infants. Braz J Med Biol Res Rev Bras Pesqui Medicas E Biol. 2023;56:e13186;
Mohammadi F, Green M, Tolsdorf E, Greffard K, Leclercq M, Bilodeau JF, et al. Industrial and Ruminant Trans-Fatty Acids-Enriched Diets Differentially Modulate the Microbiome and Fecal Metabolites in C57BL/6 Mice. Nutrients. 2023 Mar 16;15(6):1433.
- Another point is that the study hypothesis could be clearer than it is. How the "degree" of growth will be operationalized\/
- At the end of this section we can find that “This study provides longitudinal microbiome data from a large preterm infant population in an urban neonatal intensive care unit (NICU). We prospec-tively recruited patients and collected weekly stool samples to test this hypothesis.” These two sentences belong to the methods section.
METHODS
- In the section “Patients and Study Design:”
Please explain how was the invitation to participate of this study.
- In this the section “Consent” we can find that
This study was approved by the University of Texas Southwestern Medical Center and Parkland Hospital Institutional Review Boards and infants were enrolled after paren-tal written consent was given.” Please include the data of the approval and the country.
- Furthermore, in general in this section I felt some weakness in some points:
Sample:
Was there a sample calculation?
- Appropriately describe inclusion/exclusion criteria such as antibiotic use and diet).
- Data collection:
Was the fecal collection method (time, storage) standardized?
- Regarding sequencing: the region of the 16S rRNA gene is not specified.
- I see the need to explain how the laboratory quality control was not detailed.
RESULTS
- Please include the definitions for all the abbreviations used in Table 1;
- The same for table 2
DISCUSSION
- In this section, the authors comment on differences in the rates of Veillonella, Bifidobacterium, and Clostridium. I feel there is a lack of integration between these findings and their possible relationship with diet, metabolism, intestinal inflammation, and oxidative stress. Is it possible to briefly work on these points?
- Other points where I feel the discussion lacks depth are regarding antibiotics (very commonly used at birth). Do they influence the microbiome and growth? If comorbidities are present in the patients analyzed, is this a confounding factor?
- In the ABSTRACT (page 1) we can read that “Results: 116 infants were enrolled” (I really miss the number of lines!). In the first sentence of Methods we can read that “In this nested case-control study within a prospective cohort,, we enrolled 115 infants in the primary cohort less than 29-weeks gestation at birth and classified them based on postnatal growth outcomes.” The same in the first line of Results section and in the first line of the Discssion. Are the sample with 115 or 116 individuals?
- In some way, the authors state that fluctuations in Bifidobacterium, Clostridium, and Veillonella spp. may be related to "anti-microbial or inflammatory factors," but is there evidence in the study to support this stamen?
- On page 9, the authors include the limitations at the end of the discussion. I suggest a separate section.
CONCLUSION
- Although the last sentence of this section starta to buil the “future perspectives” for this study, I suggest a separate section for this. The authors can include something like:
FUTURE PERSPECTIVES
Future studies should use standardized, longitudinal sampling to better capture microbiome dynamics in preterm infants. Approaches integrating metagenomics and metabolomics are needed to clarify how specific growth rates influence growth. Future research should also isolate the impact of important clinical factors, such as nutrition, antibiotic use, and the presence of comorbidities, on microbial composition. The development of controlled clinical trials should evaluate microbiome-modulating interventions, such as targeted probiotics, prebiotics, or personalized fortification, to determine if intentional changes in the gut ecosystem can improve growth outcomes.
Reviewer 2 Report
Comments and Suggestions for Authors
The authors explored the association between gut microbiome composition and growth failure in preterm infants, with rigorous design and valuable clinical data. Some revision suggestions to improve the manuscript:
- Please add a "Sample Screening Flowchart": Convert the textual description of sample exclusion criteria (low alpha diversity, lack of longitudinal samples) into a visual flowchart or table, clearly illustrating the process from 263 initial samples to 218 valid samples.
- Please divide "microbiome analysis" into three independent subsections: Alpha/Beta Diversity Analysis, Differential Abundance Analysis, and Clustering Analysis, to avoid content confusion and improve readability.
- In the "Statistical Analysis of Microbiome Data" section, supplement the specific numerical range for "Shannon diversity and read counts below 2 standard deviations of the overall mean" to enhance reproducibility.
- Please expand variable abbreviations in the formula (e.g., “Abx status” to “Antibiotic status”, “feeding grp” to “feeding group”, “DOL_at_sample collection” to “Day of Life at sample collection” to avoid ambiguity and ensure standardized format.
- Please maintain consistent definitions of "ASVs", "taxa", and "genus-level taxa" throughout the manuscript. Provide full annotations when they first appear (e.g., ASVs: Amplicon Sequence Variants) and avoid mixing "taxon/taxa".
- Please standardize bacterial name formatting: Italicize bacterial genus and species names (e.g., Veillonella, Bifidobacterium) when they first appear, and abbreviate them thereafter (e.g., Veillonella to V.).
- Please summarize study limitations into three points: sample loss, fecal collection time deviation and feeding mode heterogeneity and provide one improvement direction for each point.
Round 2
Reviewer 1 Report
Comments and Suggestions for Authors
Dear authors,
Thank you very much for performing the corrections I suggested.
With best regards.